# Real-world outcomes of first- and second-generation tyrosine kinase inhibitors first-line in patients with epidermal growth factor receptor mutation-positive non-small cell lung cancer: A retrospective observational cohort study

Wei-Wei Ng[1], Chen-Chun Lin[1,2], Ching-Yuan Cheng[1], Jiunn-Song Jiang[1,3], Shang-Jyh Kao[1,3], Diana Yuwung Yeh[1,2]*

1 Division of Chest Medicine, Department of Internal Medicine, Shin Kong Wu-Ho-Su Memorial Hospital, Taipei, Taiwan, 2 School of Medicine, Fu Jen Catholic University, New Taipei City, Taiwan, 3 School of Respiratory Therapy, Taipei Medical University, Taipei, Taiwan

* m004917@ms.skh.org.tw

**Data Availability Statement:** All relevant data are within the manuscript and S1 Dataset.

## Abstract

The sequencing of epidermal growth factor receptor tyrosine kinase inhibitors (EGFR-TKIs) in patients with EGFR mutation-positive (EGFRm+) non-small cell lung cancer (NSCLC) remains a matter of controversy. This cohort study analyzed the overall survival (OS) and progression-free survival (PFS) of afatinib compared with erlotinib and gefitinib first-line. EGFRm+, advanced NSCLC patients treated with either afatinib, erlotinib or gefitinib were retrospectively analyzed. A total of 107 patients were included. There was no statistically significant difference in PFS among the 3 groups. In the ≥ 60 years age group, the afatinib group had longer survival compared to the gefitinib group (p = 0.01). Median OS were 19.1, 22.9, and 35.6 months for gefitinib, erlotinib, and afatinib groups, respectively, with statistical significance between the gefitinib and afatinib groups (p = 0.009). Patients on afatinib also had longer median OS than erlotinib and gefitinib pooled together (35.5 versus 21.4 months; hazard ratio = 0.54, p = 0.016), despite similar median PFS. In conclusion, afatinib is a better choice compared to gefitinib or erlotinib for EGFRm+ patients. The OS obtained with afatinib is just 3 months shorter than osimertinib in the FLAURA trial. Direct comparison studies with osimertinib are still needed to determine optimal sequencing.

## 1. Introduction

Lung cancer is the second most prevalent cancer in both men and women, also the leading cause of cancer death in the United States [1]. It is also the most common cause of cancer deaths worldwide [2]. In a cancer-based epidemiology and survival analysis, the average 5-year survival rate for NSCLC from 1973 to 2010 was only 5.6% [3]. In comparison to conventional

**Funding:** The authors received no specific funding for this work.

**Competing interests:** The authors have declared that no competing interests exist.

cytotoxic chemotherapy, epidermal growth factor receptor tyrosine kinase inhibitors (EGFR-TKIs) provide significantly improved objective response rates and progression-free survival (PFS) for patients with advanced cases of NSCLC with EGFR mutations [4–6]. The frequency of EGFR mutations varies considerably among different populations and have been reported to occur in only around 10%–15% of patients in Western countries in comparison to approximately 50% of patients in Asian countries [7]. Therefore, the optimal choice of EGFR-TKI is of particular concern in countries with high prevalence of EGFR mutations.

In patients with "common" EGFR mutations, exon 19 deletions and L858R mutation, the two first-generation EGFR-TKIs, erlotinib and gefitinib, reversibly inhibit the kinase activity of EGFR. On the other hand, the second-generation EGFR-TKIs afatinib and dacomitinib bind irreversibly to the intracellular tyrosine kinase domain, inhibiting intracellular phosphorylation and, in turn, causing cancer cell deaths. These different mechanisms of action may result in different levels of efficacy [8]. In the CTONG0901 trial, the two first-generation TKIs erlotinib and gefitinib achieved similar progression free survival (PFS) and overall survival (OS) with similar toxicity [9, 10]. In the LUX-LUNG 7 trial, on the other hand, the second-generation TKI afatinib significantly improved PFS (11.0 vs. 10.9 months, P = 0.017) compared with first-generation gefitinib. However, OS remained similar in both groups [11, 12]. The ARCHER 1050 study further showed that dacomitinib is superior to gefitinib in both PFS and OS, but patients with brain metastases were excluded from this study, which greatly limits the application of the study conclusion in many clinical settings [13, 14]. On the other hand, some patients do not carry "common" mutations but rather harbor "rare" mutations, which were not well represented in most trials of EGFR-TKIs. For instance, a study by Krawczyk *et al.* found no significant differences in the efficacy (in terms of the median OS, PFS, and treatment response) of afatinib, erlotinib, and gefitinib in patients with either common or rare EGFR mutations. However, they noted that patients with common mutations had longer PFS but similar OS on TKIs [15].

The recent phase III FLAURA study showed that the use of osimertinib, a third-generation irreversible EGFR-TKI that selectively inhibits both common EGFR mutations as well as EGFR T790M mutation, a common acquired resistance to first-generation TKIs, is associated with significantly longer PFS and OS than erlotinib and gefitinib in the first-line setting [16, 17]. Although this led to a change in clinical practice in most Western countries in the first-line treatment of NSCLC, it is worthy of note that second-generation TKIs were not used in this trial. Therefore, there is up to now still no definitive answer to the question of how to best sequence the three generations of EGFR-TKIs.

As the reports of the efficacy of the reversible EGFR-TKIs (erlotinib and gefitinib) and the irreversible EGFR-TKIs (afatinib, dacomitinib, and osimertinib) are conflicting, and the cost of each TKI varies greatly from country to country, based on each country's unique reimbursement system, not all three generations of TKIs are widely available for use in all countries. Many clinicians have had to adopt a trial-and-error approach, not necessarily able to follow published clinical guidelines. Their previous experiences with a drug often inform future clinical decisions. Therefore, to help clarify this issue for clinical practice, we conducted this hospital-based cohort study to compare the OS and PFS outcomes for afatinib, erlotinib, and gefitinib as the first-line treatment in patients with EGFR mutation-positive NSCLC.

## 2. Methods

### 2.1. Study design and patient demographics

The present study was conducted in patients with advanced NSCLC treated at a 921-bed, tertiary teaching medical center in Taipei, Taiwan. All EGFR mutation-positive patients with

NSCLC who were treated at the hospital from June 2014 to May 2019 were screened for inclusion into the study. All patients underwent a bone scan, chest computed tomography (CT) scan, and brain imaging (CT or magnetic resonance imaging) scan for staging based on the tumor, node, metastasis (TNM) classification proposed by the American Joint Committee on Cancer, 7th and later 8th edition. Stage I–IIB patients were then excluded, and only advanced-stage patients were ultimately included in the analysis. A total of 107 patients were included, divided into 3 treatment groups based on the choice of 1st line TKI used according to physician's clinical decision: 27 in the gefitinib group, 33 in the erlotinib group, and 47 in the afatinib group. Patient data were accessed and prepared by a non-author between June and August, 2019. In line with the Declaration of Helsinki, the file was fully anonymized with all identifying information removed prior to data analysis by the investigators.

Patient characteristics data were collected, including age, sex, mutation subtype, performance status, and TNM stage. All patients received either a reversible EGFR-TKI (gefitinib or erlotinib) or an irreversible EGFR-TKI (afatinib) as their first-line treatment at the discretion of the treatment providers. Data on all concurrent and subsequent treatment modalities provided to the patients, including the aforementioned initial and any subsequent TKI therapies such as third-generation osimertinib, radiation therapy, and cytotoxic chemotherapy were also collected. Based on radiographic evidence, the Response Evaluation Criteria in Solid Tumors (RECIST), version 1.1, was used to determine the occurrence of disease progression.

## 2.2. EGFR mutation analysis

Tumor tissues were obtained from the primary lung tumors or metastatic lesions of the patients for EGFR mutation analysis. Only tissue samples consisting of >80% tumor content, as evaluated via microscopy, were used for this purpose. For each sample, DNA was extracted using the QIAcube automated extractor (Qiagen, Hilden, Germany) with the QIAamp DNA Formalin-Fixed Paraffin-embedded (FFPE) Tissue Kit (Qiagen) and then eluted in ATE (QIAmp Tissue Elution) buffer (Qiagen) according to the manufacturer's instructions. The EGFR PCR Kit (EGFR RUO Kit) and the Therascreen EGFR RGQ PCR Kit (EGFR IVD Kit, Qiagen, Manchester, UK) were then used in combination with the Scorpions and amplification-refractory mutation system (ARMS) technologies to detect the presence of EGFR mutations by real-time quantitative PCR.

## 2.3. Statistical analysis

ANOVA model was used to compare the difference of continuous variables between treatment groups. Kruskal-Wallis test was used if normally distributed assumption was violated. Normal distribution was examined by the Kolmogorov-Smirnov test. Categorical variables were compared by chi-square test or the Fisher's exact test (if the expected value is smaller than 5). Survival analysis was conducted by the Kaplan-Meier method and the difference between groups was tested by the log-rank test. Cox proportional regression model was conducted to obtain the hazard ratio of the covariates.

The chi-squared test or Fisher's exact test was used to compare the categorical demographic and clinical variables of the patients, while Student's t-test was used to compare the continuous demographic and clinical variables of the patients. The PFS and OS of the patients were estimated by the Kaplan–Meier method and compared using the log-rank test. Cox proportional hazards regression analyses were also performed to ascertain the determinants of the PFS and OS of the patients. Possible determinants were selected based on prior studies of the prognostic factors for survival [18, 19]. The factors ultimately selected as possible prognostic factors were age, sex, smoking history, previous cancer status, other concurrent cancer status, family history

of lung cancer, family history of cancer in general, Eastern Cooperative Oncology Group Performance Status (ECOG PS), radiotherapy treatment status, clinical stage, nodal involvement, and EGFR mutation subtype. The analyses were performed using the Statistical Analysis System software version 9.4 (SAS Institute, Cary, North Carolina, USA). All the reported $p$-values are two-sided, and a $p$-value of less than 0.05 was considered to indicate statistical significance.

## 3. Results

### 3.1. Characteristics of patients in the two treatment groups

A total of 107 patients were included in the study. Of these patients, 47 received afatinib as their first-line treatment, 60 received erlotinib or gefitinib. (33, 27, respectively)

Comparing the pooled 1st generation TKI with the 2nd generation TKI groups, the demographic and baseline clinical characteristics of the patients in the erlotinib/gefitinib versus afatinib groups are shown in Table 1. Compared with the afatinib group, a significantly larger proportion of the patients in the erlotinib/gefitinib group were aged ≥60 years (90% versus 62%; $p = 0.001$). In addition, more patients in the erlotinib/gefitinib group had an ECOG PS ≥2 (33% versus 9%; $p = 0.002$) or were clinical stage IV (97% versus 87%; $p = 0.003$). More patients in the 1st generation TKI group harbored exon 21 mutations (62% versus 45%; $p = 0.012$), whereas more patients on afatinib had uncommon EGFR mutations (10% vs. 2%; $p = 0.012$). Furthermore, compared with the erlotinib/gefitinib group, a significantly larger proportion of the patients in the afatinib group had a family history of cancer (38% versus 18%; $p = 0.028$).

**Table 1. Demographics and baseline characteristics in patients receiving first or second generation TKI therapies.**

| Variable | Afatinib | Erlotinib/Gefitinib | *P* value |
|---|---|---|---|
| Patient number | 47 | 60 | - |
| Age ≥60 years | 29 (62) | 54 (90) | 0.001 |
| Male | 19 (40) | 27 (45) | 0.696 |
| Smoking history | 16 (34) | 17 (28) | 0.535 |
| Previous cancer | 3 (6) | 5 (8) | >0.999 |
| Other concurrent cancer | 1 (2) | 2 (3) | >0.999 |
| Family history of lung cancer | 7 (15) | 6 (10) | 0.554 |
| Family history of cancer | 18 (38) | 11 (18) | 0.028 |
| ECOG PS ≥2 | 4 (9) | 20 (33) | 0.002 |
| Ever received radiotherapy | 17 (36) | 34 (57) | 0.051 |
| Time on TKI treatment, month | 15.3 ± 13.4 | 12.3 ± 10.7 | 0.197 |
| Clinical stage | | | 0.003 |
| IIIa | 0 (0) | 2 (3) | |
| IIIb | 6 (13) | 0 (0) | |
| IV | 41 (87) | 58 (97) | |
| Nodal involvement (N ≥1) | 40 (85) | 45 (75) | 0.235 |
| EGFR mutation | | | 0.012 |
| *Wild type | 0 (0) | 1 (2) | |
| Exon 19 deletion | 16 (34) | 20 (33) | |
| Exon 21 L858R mutation | 21 (45) | 37 (62) | |
| Other EGFR mutations | 10 (21) | 2 (3) | |

Abbreviation: TKI, tyrosine kinase inhibitor; ECOG PS, Eastern Cooperative Oncology Group Performance Status; EGFR, epidermal growth factor receptor.

Data presented as frequency (percentage) or mean ± standard deviation.

*Wild type: An elderly patient with poor performance status. The patient requested TKI trial and palliative care.

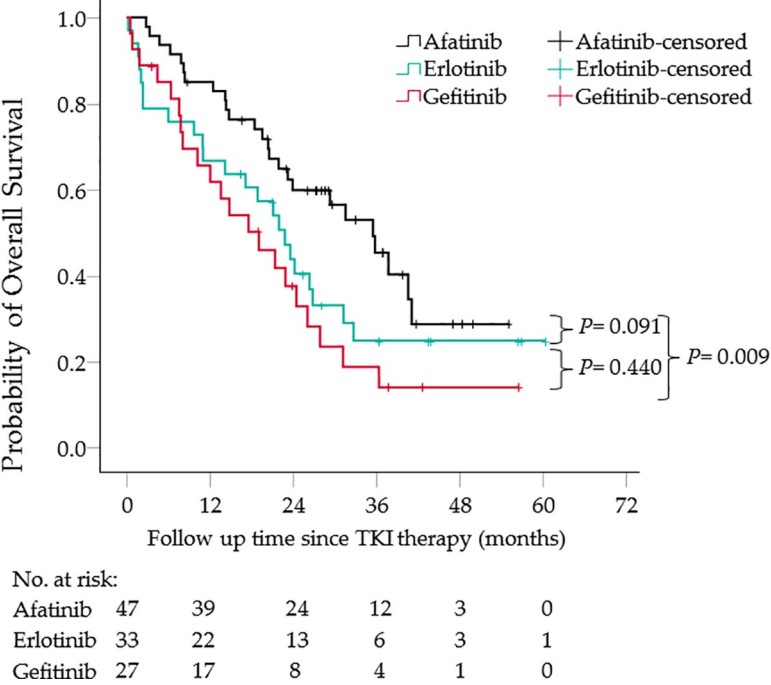

**Fig 1. Overall survival of non-small cell lung cancer patients based on first-line TKI therapies.** The median survival times are 35.6, 22.9, and 19.1 months in the afatinib, erlotinib, and gefitinib groups, respectively. Abbreviation: TKI, tyrosine kinase inhibitor.

### 3.2. PFS and OS of the patients in the two treatment groups

Fig 1 shows a higher percentage of NSCLC patients were alive at month 12, 24, and 36 on afatinib first-line as compared to gefitinib and erlotinib. Median time on treatment was 10.0 months for all 3 treatment groups. However, median overall survival time was 19.1, 22.9, and 35.6 months for gefitinib, erlotinib, and afatinib groups, respectively, with statistical significance between the gefitinib and afatinib groups ($p = 0.009$).

Pooling 1st generation TKI data together, the comparison of the OS of the patients in the afatinib and erlotinib/gefitinib groups is shown in Fig 2. The median OS of the patients in the afatinib group was significantly longer than that of the patients in the erlotinib/gefitinib groups (35.5 months versus 21.4 months; HR 0.54, $p = 0.016$). In contrast, as shown in Fig 3, the patients in the afatinib group showed no significant difference of PFS compared to those in the erlotinib/gefitinib group (12.0 months versus 13.0 months; HR 0.79, $p = 0.360$).

### 3.3. Demographic and clinical characteristics significantly associated with PFS and OS

According to the results of bivariate Cox analyses, the demographic and clinical factors that were statistically associated with higher hazard ratio of OS were having an ECOG PS $\geq 2$ ($p = 0.001$), having a wild-type EGFR mutation ($p = 0.002$), and receiving afatinib rather than erlotinib or gefitinib as a first-line treatment ($p = 0.016$). Further multivariate Cox analyses showed that the demographic and clinical factors that were significantly associated with higher hazard ratio of OS were being male ($p = 0.039$), having an ECOG PS $\geq 2$ ($p = 0.005$), and having a wild-type EGFR mutation ($p = 0.033$) (Table 2). In terms of PFS, the bivariate Cox analyses did not reveal any significant demographic or clinical factors (Table 3).

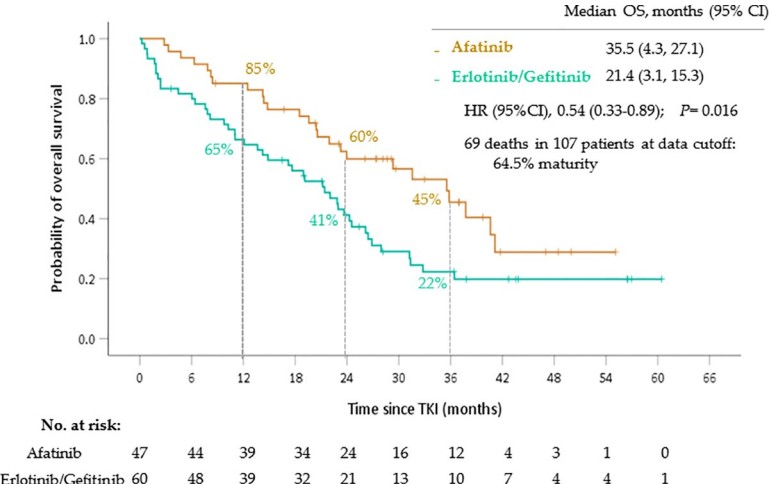

**Fig 2. Overall survival of non-small cell lung cancer patients based on first-line TKI therapies.** Abbreviation: TKI, tyrosine kinase inhibitor.

## 4. Discussion

Potential variations in efficacy resulting from different treatment sequencing options have significant practical implications for patients and clinicians alike. Previous reports of the survival benefits of the second-generation irreversible EGFR-TKIs afatinib and dacomitinib in comparison to those of erlotinib and gefitinib are conflicting [9, 11–15]. In the recent phase III FLAURA clinical trial, previously untreated, common EGFR mutation-positive NSCLC patients who received osimertinib were compared with those who received either erlotinib or gefitinib. The FLAURA data showed that those treated with osimertinib had both a significantly longer median PFS (18.9 months versus 10.2 months; HR 0.46) and a significantly longer OS (38.6 months versus 31.8 months; HR 0.80) than those treated with either of the

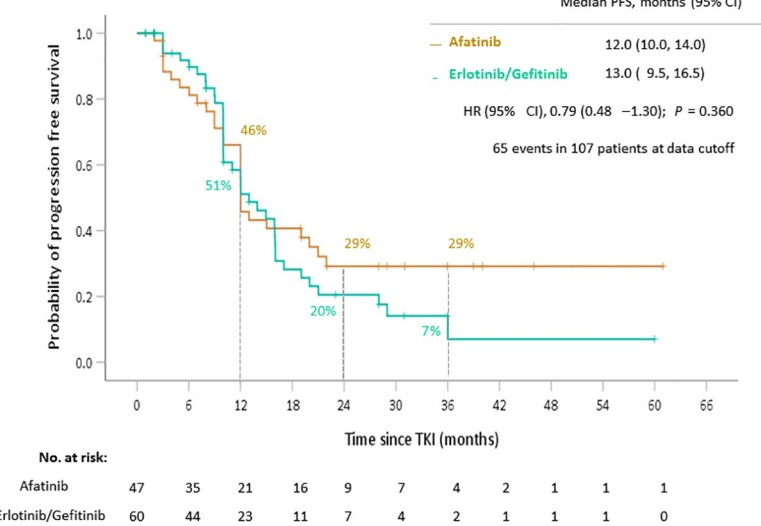

**Fig 3. Progression free survival of non-small cell lung cancer patients based on first-line TKI therapies.** Abbreviation: TKI, tyrosine kinase inhibitor.

**Table 2. Bivariable and multivariable analyses of the association between clinical characteristics and overall survival.**

| Variable | Bivariable analysis | | Multivariable analysis | |
|---|---|---|---|---|
| | HR (95% CI) | *P* | HR (95% CI) | *P* |
| Age ≥60 years | 0.99 (0.57–1.73) | 0.965 | | |
| Male | 1.55 (0.97–2.49) | 0.070 | 1.72 (1.03–2.90) | 0.039 |
| Smoking history | 1.10 (0.66–1.83) | 0.713 | | |
| Previous cancer | 1.16 (0.47–2.88) | 0.754 | | |
| Other concurrent cancer | 2.42 (0.75–7.80) | 0.140 | 1.54 (0.45–5.20) | 0.489 |
| Family history of lung cancer | 1.03 (0.51–2.09) | 0.926 | | |
| Family history of cancer | 0.77 (0.45–1.32) | 0.340 | | |
| ECOG PS ≥2 | 2.46 (1.47–4.11) | 0.001 | 2.35 (1.30–4.27) | 0.005 |
| Ever received radiotherapy | 1.44 (0.89–2.31) | 0.134 | 1.20 (0.72–2.00) | 0.480 |
| Clinical stage IV vs. III | 0.93 (0.37–2.31) | 0.875 | | |
| Nodal involvement (N ≥1) | 0.95 (0.54–1.66) | 0.848 | | |
| EGFR mutation | | | | |
| Wild type | 34.3 (3.5–337) | 0.002 | 13.17 (1.23–141) | 0.033 |
| Exon 19 deletion | Reference | | | |
| Exon 21 L858R mutation | 1.02 (0.61–1.69) | 0.951 | 0.88 (0.51–1.51) | 0.646 |
| Other EGFR mutations | 0.78 (0.32–1.91) | 0.593 | 0.91 (0.37–2.25) | 0.841 |
| Afatinib vs. Erlotinib/Gefitinib | 0.54 (0.33–0.89) | 0.016 | 1.39 (0.78–2.47) | 0.266 |

Abbreviation: HR, hazard ratio; CI, confidence interval; TKI, tyrosine kinase inhibitor; ECOG PS, Eastern Cooperative Oncology Group Performance Status; EGFR, epidermal growth factor receptor.

**Table 3. Bivariable and multivariable analyses of the association between clinical characteristics and progression free survival.**

| Variable | Bivariable analysis | | Multivariable analysis | |
|---|---|---|---|---|
| | HR (95% CI) | *P* | HR (95% CI) | *P* |
| Age ≥60 years | 0.81 (0.47–1.39) | 0.442 | | |
| Male | 1.55 (0.94–2.54) | 0.085 | 1.57 (0.95–2.61) | 0.080 |
| Smoking history | 1.01 (0.59–1.71) | 0.981 | | |
| Previous cancer | 0.68 (0.21–2.17) | 0.513 | | |
| Other concurrent cancer | 2.26 (0.70–7.27) | 0.171 | | |
| Family history of lung cancer | 1.11 (0.56–2.17) | 0.773 | | |
| Family history of cancer | 0.88 (0.51–1.52) | 0.644 | | |
| ECOG PS ≥2 | 0.74 (0.37–1.51) | 0.411 | | |
| Ever received radiotherapy | 1.67 (1.02–2.73) | 0.042 | 1.14 (0.68–1.88) | 0.622 |
| Clinical stage IV vs. III | 3.39 (0.83–13.90) | 0.090 | 1.89 (0.45–7.93) | 0.383 |
| Nodal involvement (N ≥1) | 1.65 (0.78–3.45) | 0.188 | | |
| EGFR mutation | 0.81 (0.47–1.39) | 0.442 | | |
| Wild type | NA | NA | | |
| Exon 19 deletion | Reference | | Reference | |
| Exon 21 L858R mutation | 0.79 (0.47–1.34) | 0.381 | | |
| Other EGFR mutations | 1.03 (0.46–2.31) | 0.937 | | |
| Afatinib vs. Erlotinib/Gefitinib | 0.79 (0.48–1.30) | 0.360 | | |

Abbreviation: HR, hazard ratio; CI, confidence interval; TKI, tyrosine kinase inhibitor; ECOG PS, Eastern Cooperative Oncology Group Performance Status; EGFR, epidermal growth factor receptor; NA, not applicable.

first-generation reversible EGFR-TKIs [16, 17]. Several studies of osimertinib either in real-world European populations or Asian populations have also revealed good responses to those with T790M mutations, albeit costly [20, 21]. However, the patients in the FLAURA trial did not have second-generation TKIs as a treatment option either first-line or post-first-line, and studies comparing osimertinib and second-generation TKIs such as afatinib or dacominitib are still lacking at present time. It is well-known that resistance patterns differ when the same drug is used at different time points in the line-up of therapies depending on what medications the patient has been exposed to prior [22]. This will in turn have an impact on the available choice, response, and duration of subsequent therapies. In our study, both first- and second-generation TKIs had similar PFS yet very different OS. Although the differences in performance status and age of the patients most likely contributed to the difference in OS between the afatnib and the gefitnib/erlotinib groups, it is also conceivable that the longer OS in the afatinib group may be the effect of different resistance mechanisms emerging from treatment with the two different categories of TKIs. With their corresponding range of sensitive mutations, they lead to diverging landscapes of available treatment options and variable response patterns to therapies down the line. Therefore, as second- and third-generation TKIs have not been studied together in large-scale clinical trials, the role of first-line second-generation TKIs in present day treatment of NSCLC has not really been clarified.

The median OS of the afatinib group in the present real-world study (35.6 months) is comparable to that obtained by osimertinib in the FLAURA study (38.6 months) under trial settings [17], considering that most of the patients included in our study did not have a chance to use osimertinib as a subsequent treatment option as osimertinib was not reimbursed by the National Health Insurance in Taiwan during most of the study period. Only 2 patients in the gefitinib group and 1 each in the erlotinib and afatinib groups used osimertinib in later-line treatment. In addition, if we compare our real-world PFS and OS to previously published afatinib clinical trial (LUX-LUNG 7) [11], our PFS are slightly longer than those in the trials in both the afatinib and the erlotinib/gefitinib groups despite worse performance status in both groups. Our OS in the 1st-generation TKI arm is only 3 mo shorter than the LUX-LUNG 7 OS (21.4 mo vs. 24.5 mo), with 33% of PS $\geq$ 2 patients in our study; and 7 mo longer compared to LUX-LUNG 7 in the afatinib arm (35.6 mo vs. 27.9 mo) despite including 9% of PS $\geq$ 2 patients. Such patients with poor performance status were excluded in LUX-LUNG 7 and FLAURA trials. Therefore, our study results indicate that the efficacy of first-line afatinib may not be inferior to that of osimertinib for some EGFR mutation-positive NSCLC patients. Given the greater cost of osimertinib, afatinib may be a reasonable choice for first-line treatment in EGFR mutation-positive NSCLC especially in areas where osimertinib is too costly. For example, under the Taiwan National Health Insurance reimbursement policy, afatinib currently costs $1,392 USD/mo compared with osimertinib at $5,649 USD/mo. According to the cost-effectiveness threshold criteria of the World Health Organization, Aguiar et al. reported that osimertinib is not, in fact, a cost-effective first-line therapy for advanced EGFR mutation-positive NSCLC [20]. A Dutch study also reached a similar conclusion [21].

From a medical standpoint, the presence of uncommon EGFR mutations is another consideration when deciding between the second- and third-generation TKIs. Both in vitro studies [23] and in vivo reports have shown that afatinib can inhibit some uncommon EGFR mutations which osimertinib has poor activities against. A previous report has revealed that EGFR L718V mutation mediates resistance to osimertinib, but retains sensitivity to afatinib, likewise with the EGFR L718Q mutation [24, 25]. There have been several other reports of incidences where afatinib has been effective in cases resistant to Osimertinib [25–28]. We also postulate that the use of afatinib may have affected the emergence of resistant mutations, perhaps contributing to a longer OS in our study than the LUX-LUNG 7 trial, as our afatinib group

included 21% of patients with uncommon mutations. As the treatment of cancer has become more personalized and targeted at the molecular level, treatment choices based on general guidelines need to be weighed carefully while considering the individual patient's mutation profile.

Some limitations of the present study should be noted. Because of the retrospective study design and analysis of data from a single hospital, the two treatment groups were significantly different in terms of some baseline clinical and demographic characteristics. The patients in the erlotinib/gefitinib group were significantly older and had worse ECOG PS than those in the afatinib group. However, further analysis using the Cox proportional hazard model to adjust for such potential confounding factors showed that afatinib treatment remained an independent predictor of better OS in this patient population.

## 5. Conclusions

In conclusion, first-line afatinib in EGFR mutation-positive NSCLC patients is associated with longer overall survival despite similar median time on treatment compared to erlotinib and gefitinib. The OS of afatinib in real-world settings is only 3 months shorter than osimertinib under trial conditions. Therefore, afatinib may be considered an alternative to osimertinib for first-line treatment of EGFR-positive NSCLC especially if medical cost is a concern or if patients have uncommon EGFR mutations. Further large-scale prospective studies are required to more comprehensively compare the effectiveness of afatinib and osimertinib in advanced EGFR mutation-positive NSCLC.

## Supporting information

**S1 Dataset.**
(CSV)

## Author Contributions

**Conceptualization:** Wei-Wei Ng, Diana Yuwung Yeh.

**Data curation:** Wei-Wei Ng, Diana Yuwung Yeh.

**Formal analysis:** Wei-Wei Ng.

**Investigation:** Wei-Wei Ng.

**Methodology:** Wei-Wei Ng.

**Project administration:** Wei-Wei Ng.

**Resources:** Wei-Wei Ng, Chen-Chun Lin, Jiunn-Song Jiang, Shang-Jyh Kao, Diana Yuwung Yeh.

**Software:** Wei-Wei Ng.

**Supervision:** Diana Yuwung Yeh.

**Validation:** Wei-Wei Ng, Ching-Yuan Cheng, Diana Yuwung Yeh.

**Visualization:** Wei-Wei Ng.

**Writing – original draft:** Wei-Wei Ng.

**Writing – review & editing:** Diana Yuwung Yeh.

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
