## [Decision Letter · Decision Letter 0]

18 May 2021

PONE-D-20-37227

Real-world outcomes of first- and second-generation tyrosine kinase inhibitors first-line in patients with epidermal growth factor receptor mutation-positive non-small cell lung cancer: a retrospective observational cohort study

PLOS ONE

Dear Dr. Diana Yeh,

Thank you for submitting your manuscript to PLOS ONE. After careful consideration, we feel that it has merit but does not fully meet PLOS ONE’s publication criteria as it currently stands. Therefore, we invite you to submit a revised version of the manuscript that addresses the points raised during the review process.

We look forward to receiving your revised manuscript.

Kind regards,

Ramon Andrade De Mello, MD, PhD, FACP

Academic Editor

PLOS ONE

Journal Requirements:

3. In the ethics statement in the manuscript and in the online submission form, please provide additional information about the patient records used in your retrospective study, including: a) whether all data were fully anonymized before you accessed them; b) the date range (month and year) during which patients' medical records were accessed. If the ethics committee waived the need for informed consent, or patients provided informed written consent to have data from their medical records used in research, please include this information.

4. Please amend the manuscript submission data (via Edit Submission) to delete author Wei Wei Ng. We note this author is added twice.

5. We note you have included a table to which you do not refer in the text of your manuscript. Please ensure that you refer to Table 3 in your text; if accepted, production will need this reference to link the reader to the Table.

Reviewers' comments:

Reviewer's Responses to Questions

**Comments to the Author**

1. Is the manuscript technically sound, and do the data support the conclusions?

Reviewer #1: Partly

Reviewer #2: Yes

2. Has the statistical analysis been performed appropriately and rigorously? 

Reviewer #1: N/A

Reviewer #2: Yes

3. Have the authors made all data underlying the findings in their manuscript fully available?

Reviewer #1: No

Reviewer #2: Yes

4. Is the manuscript presented in an intelligible fashion and written in standard English?

Reviewer #1: Yes

Reviewer #2: Yes

5. Review Comments to the Author

Reviewer #1: To the critical analysis of the article, I suggest the evaluation in the following topics:

- Explain which physicians criteria were used for the allocation of patients to groups

- Can not the heterogeneity between the groups be considered a selection bias resulting from the decision of the responsible physician?

- Regarding the analysis of the EGFR mutation research, elucidate the validity of the methodology employed

- Discuss why are the primary end points (OS and PFS) related to the effectiveness of alfatinib and not the heterogeneity of the groups? Group B was allocated to patients aged> 60 years, in a more advanced stage of the disease (97% stage IV), with ECOG PS> 2 (33%).

- In line 194, 198, 219, 224 and 226, references 20-28 are cited but are not included in the article

Reviewer #2: ONE LIMITATION OF THE STUDY WAS THE POPULATION OF THE ERLOTINIB/GEFITINIB GROUP THAT WAS OLDER AND WITH POOR PERFORMANCE STATUS THAN AFATINIB GROUP. IT WOULD BE INTERESTING TO DO A SUBGROUP ANALYSIS WITH ONLY THESE PATIENTS TO SEE THE BENEFIT OF THE DRUGS AND ADD THEIR OS AND PFS IN THE STUDY. I BELIEVE IT IS NOT NECESSARY TO DESCRIBE THE COST EFFECTIVENESS OF THE MEDICATIONS ONCE IT IS NOT AN END POINT IN THIS STUDY

6. PLOS authors have the option to publish the peer review history of their article (what does this mean?). If published, this will include your full peer review and any attached files.

Reviewer #1: No

Reviewer #2: No

---

## [Author Response · Author response to Decision Letter 0]

30 May 2021

Response to Reviewers

Reviewer #1: To the critical analysis of the article, I suggest the evaluation in the following topics:

- Explain which physicians criteria were used for the allocation of patients to groups

Answer: In Taiwan, all three medications (gefitinib, erlotinib, and afatinib) are reimbursed by the National Health Insurance, available to lung cancer patients with EGFR mutation 1st line without further restrictions. The choice of drug is usually a shared decision between the physician and the patient/family based on potential side effects vs. perceived potency of the drug. As this is a retrospective chart review, we set no “criteria” for the choice of drug. It was entirely the treating physician’s decision after consultation with the patient/family. A key factor is usually the severity of tyrosine kinase inhibitor induced diarrhea. Afatinib is known to be associated with more severe diarrhea.

- Can not the heterogeneity between the groups be considered a selection bias resulting from the decision of the responsible physician?

Answer: Following the 1st question, yes and no. The National Health Insurance allows a patient to switch from afatinib to gefitinib or erlotinib if he cannot tolerate the side effects. The treating physician can “dissuade” the patient/family from choosing afatinib at the outset, but he can also let the patient try it for 1 week or 2 and THEN switch if it is not tolerated, and indeed this is what treating physicians usually do, as afatinib is only reimbursed for 1st line use while the other two can be used in subsequent lines as well. It is fair to say that if a patient is frail and cannot tolerate the side effects of afatinib, he would likely have “self-selected” to be in the gefitinib/erlotinib group after a trial period.

- Regarding the analysis of the EGFR mutation research, elucidate the validity of the methodology employed

Answer: In our facility during this period, EGFR testing was done by the EGFR PCR Kit (EGFR RUO Kit) and the Therascreen EGFR RGQ PCR Kit (EGFR IVD Kit, Qiagen, Manchester, UK), used in combination with the Scorpions and amplification-refractory mutation system (ARMS) technologies to detect the presence of EGFR mutations by real-time quantitative PCR. This is a widely available and well-validated method. This part is explained in the manuscript under Methods 2.2.

- Discuss why are the primary end points (OS and PFS) related to the effectiveness of alfatinib and not the heterogeneity of the groups? Group B was allocated to patients aged> 60 years, in a more advanced stage of the disease (97% stage IV), with ECOG PS> 2 (33%).

Answer: Indeed, we cannot rule out that possibility. The differences in PS and age most likely also contributed to the difference in OS between the afatnib and the gefitnib/erlotinib groups. The PFS, however, were SIMILAR in both groups despite the PS and the age difference. As this is not a randomized, controlled, head-to-head trial, we not only compare the two groups as best we can; we also compare them to historical trial data. If we compare our real-world PFS and OS to previously published clinical trials (Lux-Lung 7), our PFS are slightly longer than those in the trials in both the afatinib and the erlotinib/gefitinib groups despite worse performance status in both groups (the presence of patients with PS ≥ 2 in our study) and the inclusion of uncommon EGFR mutations, which usually portends worse survival. Our OS is only 3 mo shorter than the Lux-Lung 7 1st-generation TKI arm (21.4 mo vs. 24.5 mo) with 33% of PS ≥ 2 patients, and 7 mo LONGER in the afatinib arm (35.6 mo vs. 27.9 mo) despite including 9% of PS ≥ 2 patients. Given that both groups of our patients have worse PS and less favorable mutations but still managed to achieve better survival than the trial data, we feel that in real-world settings, these agents still have their role in treating lung cancer patients with EGFR mutations 1st line, especially for patients who do not fit the typical clinical trial demographics. We also postulate that the use of afatinib may have affected the emergence of resistant mutations, perhaps contributing to a longer OS as this group included 21% of patients with uncommon mutations. The clinical validation of this hypothesis will require further research. We have revised the Discussion section slightly to explain this part in more details.

- In line 194, 198, 219, 224 and 226, references 20-28 are cited but are not included in the article

Answer: Sorry! Somehow they were deleted by accident. This oversight has been corrected. The Reference section has been re-checked. Thank you for pointing that out.

Reviewer #2: ONE LIMITATION OF THE STUDY WAS THE POPULATION OF THE ERLOTINIB/GEFITINIB GROUP THAT WAS OLDER AND WITH POOR PERFORMANCE STATUS THAN AFATINIB GROUP. IT WOULD BE INTERESTING TO DO A SUBGROUP ANALYSIS WITH ONLY THESE PATIENTS TO SEE THE BENEFIT OF THE DRUGS AND ADD THEIR OS AND PFS IN THE STUDY. I BELIEVE IT IS NOT NECESSARY TO DESCRIBE THE COST EFFECTIVENESS OF THE MEDICATIONS ONCE IT IS NOT AN END POINT IN THIS STUDY

Answer: Thank you for the insightful comment. We will certainly consider that when we have accumulated more patients in our database. We discussed the cost effectiveness of these therapeutics even though it is not the main point in this study because this is an important consideration in countries with one payer insurance systems. Many countries delay the introduction of certain high-cost therapeutics out of fiscal considerations. Patients end up having to pay out of pocket. Many cannot afford it.

---

## [Editor Report · Decision Letter 1]

3 Jun 2021

Real-world outcomes of first- and second-generation tyrosine kinase inhibitors first-line in patients with epidermal growth factor receptor mutation-positive non-small cell lung cancer: a retrospective observational cohort study

PONE-D-20-37227R1

Dear Dr. Diana Yeh,

We’re pleased to inform you that your manuscript has been judged scientifically suitable for publication and will be formally accepted for publication once it meets all outstanding technical requirements.

Kind regards,

Ramon Andrade De Mello, MD, PhD, FACP

Academic Editor

PLOS ONE

---

## [Editor Report · Acceptance letter]

16 Jun 2021

PONE-D-20-37227R1 

Real-world outcomes of first- and second-generation tyrosine kinase inhibitors first-line in patients with epidermal growth factor receptor mutation-positive non-small cell lung cancer: a retrospective observational cohort study 

Dear Dr. Yeh:

I'm pleased to inform you that your manuscript has been deemed suitable for publication in PLOS ONE. Congratulations! Your manuscript is now with our production department. 

Kind regards, 

on behalf of

Professor Ramon Andrade De Mello 

Academic Editor

PLOS ONE